Manuscript prepared for Clim. Past
with version 2015/09/17 7.94 Copernicus papers of the LATEX class copernicus.cls.
Date: 13 June 2016

# Could the Pliocene constrain the Equilibrium Climate Sensitivity?

J.C. Hargreaves[1] and J.D. Annan[1]

[1]BlueSkiesResearch.org.uk , The Old Chapel, Albert Hill, Settle , BD24 9HE

*Correspondence to:* J.C. Hargreaves (jules@blueskiesresearch.org.uk)

**Abstract.** The mid-Pliocene Warm Period (mPWP) is the most recent interval in which atmospheric carbon dioxide was substantially higher than in modern pre-industrial times. It is, therefore, a potentially valuable target for testing the ability of climate models to simulate climates warmer than the pre-industrial state. The recent Pliocene model inter-comparison Project (PlioMIP) presented boundary conditions for the mPWP, and a protocol for climate model experiments. Here we analyse results from the PlioMIP and, for the first time, discuss the potential for this interval to usefully constrain the equilibrium climate sensitivity. We observe a correlation in the ensemble between their tropical temperature anomalies at the mPWP, and their equilibrium sensitivities. If the real world is assumed to also obey this relationship, then the reconstructed tropical temperature anomaly at the mPWP can in principle generate a constraint on the true sensitivity. Directly applying this methodology using available data yields a range for the equilibrium sensitivity of 1.9–3.7°C, but there are considerable additional uncertainties surrounding the analysis which are not included in this estimate. We consider the extent to which these uncertainties may be better quantified and perhaps lessened in the next few years.

## 1 Introduction

One important motivation for the study of paleoclimates is that they may provide information as to how the climate will change in the future. The temperature response to changes in radiative forcing provides one simple way to summarise this through the equilibrium or Charney climate sensitivity, $S$. This is defined as the equilibrium response of the globally averaged surface air temperature (SAT) to a doubling of atmospheric $CO_2$ concentration. As a key measure of climate changes, this is one of the principal parameters by which we understand and interpret climate system behaviours.

There is evidence of both warmer and colder climates in the past. As we look increasingly further back in time, the evidence available in the paleorecord generally becomes both more sparse and less certain, and for this reason it is usually advantageous to focus research on the more recent past where possible. The most recent periods with climates that are substantially different to the present on the global scale have typically been colder than present with large ice sheets over northern continents

(i.e., the ice ages). While the Last Glacial Maximum (LGM, 21ka BP) has been extensively studied, it is challenging to draw inferences from colder climates regarding our warmer future, in part because of the ice sheets that strongly affect the climate system over large areas of the Northern Hemisphere and which may combine nonlinearly with other forcings. Thus increased attention has recently been given to warmer periods (Lunt et al., 2013). These are generally more distant in time, and data are less certain, but the inference from past to future is potentially more robust as the past climate is warmer than present and more similar to what we expect to see in the future, with for example changes in ice sheets being relatively small. It is this inference that the current paper explores. We focus on the mid-Pliocene warm period (mPWP), 2.97–3.29 million years before the present, as this represents the most recent time that the atmospheric $CO_2$ level was substantially higher than in pre-industrial times and substantial effort has been made to collect data from this interval (Dowsett et al., 2009), which also suggests that the mPWP climate was warmer than the pre-industrial.

Researchers have previously explored the mPWP as a constraint on the Earth System Sensitivity (ESS), a broader concept than Equilibrium Climate Sensitivity $S$ which also considers the longer-term feedbacks involved in the evolution of the ice sheets, and also changes in vegetation (Lunt et al., 2010). The aim of this paper is to explore the possibility that the mPWP may inform directly on the equilibrium climate sensitivity in which only the physical feedbacks of ocean, atmosphere and sea ice are considered. The methodology adopted is similar to that of Hargreaves et al. (2012) who used simulations of the LGM. The underlying hypothesis is that the models with stronger response to past radiative forcing changes, will also have a stronger response to current and future radiative forcing changes. If this hypothesis is correct, it should be evident as a relationship (most simply, a linear correlation) between past and future warming across the ensemble. If a correlation is indeed observed, then data relating to the past warming should, in principle, be able to help constrain the future (Schmidt et al., 2014a).

In the next section we consider some technical aspects of the method employed in the context of previous work on the LGM. Then in the Analysis section we introduce the models, the results from the correlation, the data, and then the estimate of equilibrium climate sensitivity. In the following section we test the sensitivity of the result to uncertainties inherent in the calculation. Finally we discuss the results and the prospect for decreasing some uncertainties in the future.

## 2 Methodology

The basic premise underlying the analysis is that, if there is a relationship between past and future behaviour across an ensemble of models, then (under the assumption that reality also obeys this relationship) observations of the past can be used to determine which of the models, and thus which future outcomes, are more reliable. Boé et al. (2009) provides an example of this idea (which is

sometimes referred to as an "emergent constraint"), using recent changes in sea ice extent to predict the future decline.

In principle it is possible to exhaustively explore an ensemble of climate model simulations for all possible relationships between past and future climate changes in variables of interest. For any cases where such a relationship is found (and for which we can also estimate the past change through some observation or climate reconstruction) we could in theory generate a forecast of the future change. However, there is a strong risk that this data mining process will generate spurious results that will not be borne out in reality (Caldwell et al., 2014). More immediately, the relationship may not be supported by the next generation of climate models (Fasullo and Trenberth, 2012; Grise et al., 2015). Thus, it is also important to ensure that the relationship is a physically meaningful one that represents our understanding of the climate system, which reduces the likelihood that it is merely a spurious correlation arising through chance.

The methodology employed here is essentially the same as that used in Hargreaves et al. (2012). In that work, the authors found a significant correlation in the ensemble from PMIP2 (the second phase of the Paleoclimate Modelling Inter-comparison Project Braconnot et al. (2007)) between the modelled cooling in the tropical ocean during the LGM, and the equilibrium climate sensitivity. This is a physically plausible result, as the temperature anomaly in the tropical region at the LGM is expected to be strongly dominated by greenhouse gas (GHG) forcing, and the tropical region (representing 50% of the globe) contributes substantially to global mean temperature changes. Furthermore, the response to $CO_2$ forcing is, at least in models, close to linear over this range of positive and negative forcing changes. Based on the correlation that Hargreaves et al. (2012) obtained, they created a simple linear regression model which used the LGM tropical temperature anomaly to predict the equilibrium sensitivity, and applied this to estimate the Earth's equilibrium sensitivity from a reconstruction of the actual LGM tropical temperature anomaly. However, it must also be noted that the correlation for the LGM, although statistically significant, was not overwhelmingly strong. Moreover, the PMIP3 ensemble gave much more equivocal results (Harrison et al., 2015; Hopcroft and Valdes, 2015). Thus, it remains challenging to use the LGM to quantitatively constrain $S$.

One issue that Hargreaves et al. (2012) did not discuss, was whether the relationship should be considered in terms of $S$ regressed on the tropical paleoclimate temperature anomaly, or vice versa. The two approaches rest on different assumptions regarding the regression residuals, and therefore can be expected to generate different results (Draper et al., 1966). An intermediate method which allows for residuals in both variables, such as total least squares, could also in principle be applied, which will give intermediate results depending on the relative weighting on the residuals on the two axes. The data as used are essentially the results of deterministic calculations and do not contain significant 'errors' as such, and although experimental protocols lead to some uncertainties in the calculated values, we do not believe that these are responsible for the residuals in the linear fit.

Therefore, the question is one of whether we can consider the residuals to be independent of one or other of the data sets.

The implicit assumption for the choice made in Hargreaves et al. (2012), of regressing $S$ on LGM tropical temperature, is that the deviations in sensitivity value from the regression line are predominantly due to factors which can be considered independent of the LGM tropical response. Further consideration supports this choice for both the LGM and mPWP according to the following argument. Uncertainty in the equilibrium sensitivity $S$ can be decomposed into various physical processes and feedbacks, including most significantly the response of clouds at both low and high latitudes, snow and ice albedo feedbacks which both act mainly at high latitudes, and other smaller factors. It is, therefore, not surprising that looking at the response in the tropics alone cannot give a precise indication of $S$, as it does not inform on the high-latitude feedbacks. This would remain the case even if we were to analyse the tropical response of a doubled $CO_2$ integration, and can be equivalently understood as different models having different degrees of polar amplification of warming. The uncertainties arising from these additional factors at higher latitudes are conceptually independent of the tropical response, as they arise from fundamentally different physical processes, and thus we can reasonably try to apply the linear model

$$S = \alpha T_{trop} + C + \epsilon$$

where $T_{trop}$ is here the tropical temperature response, $\alpha$ and $C$ are *a priori* unknown constants and the error term $\epsilon$ includes the uncertainties due to factors such as the uncertainties in the high latitude feedbacks discussed above.

In the inverse regression, where we would try to use the equilibrium sensitivity to predict tropical temperature changes, the uncertainties over and above the underlying linear relationship would have to be assumed independent of $S$, which does not seem so conceptually appropriate. That is, in applying the inverse regression

$$T_{trop} = \alpha S + C + \epsilon$$

we would have to consider the residuals $\epsilon$ here to be independent of the sensitivity $S$, even though the sensitivity is by definition the overall effect of all feedbacks in all regions.

## 3 Analysis

### 3.1 The models

The Pliocene Model Inter-comparison Project (PlioMIP, Haywood et al. (2010, 2011)) has presented boundary conditions in order for climate models to simulate the mPWP. This was not a true "time

slice" experiment such as the LGM simulations, which represented the climatic average over an interval of 19–23ka BP. The much longer mid-Pliocene interval contained multiple ice age cycles, and the mPWP experiments were designed to represent a typical or average interglacial within this period. There were two experiments conducted in PlioMIP. Experiment 1 (Haywood et al., 2010) used atmosphere-only climate models, with the sea surface temperature boundary condition prescribed from a reconstruction which is discussed further in the next section. For these simulations, we expect the SAT anomaly to be tightly constrained by the imposed boundary conditions (especially over the ocean) and therefore to bear little relationship with the model's sensitivity. The model results bear this out, and thus we do not consider these simulations further. There were 10 models that performed Experiment 2, in which coupled atmosphere-ocean general circulation models were forced with a suite of boundary conditions including a land-sea mask, topography, ice-sheet, vegetation, and green house gas concentration (see Haywood et al. (2011) for details). For these models, we expect their mPWP simulations (and in particular their SAT response) to be related to their climate sensitivities, since the greenhouse gas boundary condition forms a large part of the total forcing. In order to relate past to future, however, we can only use models for which both the mPWP simulation results and an estimate of the model's sensitivity is available. The GENISIS model is mentioned in Haywood et al. (2013) but results are not available in the PlioMIP database, so this condition reduces the ensemble to the 9 models which are listed in Table 1. The ensemble size, while smaller than might be hoped for given that more than 20 models contributed to the Climate Model Intercomparison Project, CMIP5, is of very similar size to that available for the LGM, where there are 8 models in PMIP2 and 9 models in PMIP3 satisfying equivalent criteria. For most models, the values of equilibrium climate sensitivity are taken from the estimates published in Table 1 of Haywood et al. (2013). The relevant sensitivity value for the FGOALS model was not included in that paper, but has since been published elsewhere (Zheng et al., 2013).

| Model | Reference | S (K) |
|---|---|---|
| COSMOS | Stepanek and Lohmann (2012) | 4.1 |
| CCSM4 | Rosenbloom et al. (2013) | 3.2 |
| FGOALS-g2 | Zheng et al. (2013) | 3.7 [1] |
| GISS ModelE2-R | Chandler et al. (2013) | 2.8 |
| HadCM3 | Bragg et al. (2012) | 3.1 |
| IPSLCM5A | Contoux et al. (2012) | 3.4 |
| MIROC4m | Chan et al. (2011) | 4.05 |
| MRI-CGCM2.3 | Kamae and Ueda (2012) | 3.2 |
| NorESM-L | Zhang et al. (2012) | 3.1 |

**Table 1.** Model data used in the analysis. [1] (Zheng et al., 2013), all other values taken from Haywood et al. (2013)

Raised atmospheric $CO_2$ is one of the more significant changes in boundary conditions for the mPWP, with other forcings contributing less than half as much again (Lunt et al., 2010), so it seems *a priori* reasonable to hope for a correlation in the climate model ensemble between their equilibrium sensitivities and their SAT changes at the mPWP. However, the other boundary condition changes are not negligible and if the models respond very differently to these (or nonlinearly to combinations of forcings) then a correlation between global SAT anomaly at mPWP and equilibrium sensitivity may not be observed.

## 3.2 Correlation Analysis

As a first investigation, we tested for a correlation between global SAT anomaly in the mPWP simulations, vs $S$. As the left plot of Figure 1 shows, there is a weak correlation between these two variables of 0.59, but this does not reach the 95% significance threshold of 0.67. As in Hargreaves et al. (2007) and Hargreaves et al. (2012), we anticipate that the relationship between $S$ and paleoclimate changes is likely to be stronger if we focus on the tropics for the paleosimulations, since this will reduce the influence of ice sheet and vegetation changes. This is borne out by the right hand panels of Figure 1 which show both the correlations for both pointwise (on a 10 degree grid), and zonally-averaged paleosimulations versus $S$. The model ensemble exhibits a strong correlation between mPWP tropical SAT anomaly and $S$. Integrating over the entire tropical region 30S–30N, the correlation between tropical mPWP SAT anomaly and climate sensitivity is 0.73.

## 3.3 The data

While the small ensemble gives us cause for concern (compare Hargreaves et al. (2012) with Schmidt et al. (2014a) and Hopcroft and Valdes (2015)) we proceed under the assumption that it is informative regarding the real climate system. In order to test the potential for constraining the climate system using information from the mPWP, we need an estimate of typical tropical temperatures during this period. As our reconstruction of mPWP temperatures we use the PRISM3 SST anomaly field which was presented by Dowsett et al. (2009). This is based on the PRISM3D data set, firstly processed into warm peak averages (to represent typical interglacial conditions within the "time slab" of interest) for both February and August, then converted to anomalies relative to modern conditions and finally smoothed and interpolated in both time and space into complete SST anomaly fields under the assumption that SST patterns were similar to the present. More sophisticated methods could in principle be used for the SST reconstruction, such as were presented by Zammit-Mangion et al. (2014) and Bragg (2014), but this is outside the scope of this paper. We use the average of these data fields for our analysis (equivalently, the annual average of the monthly fields that were generated for PlioMIP Experiment 1).

We have also directly investigated the data points, in the form of the annual mean temperature anomaly estimates which were provided by Dowsett et al. (2009). The locations of these data can

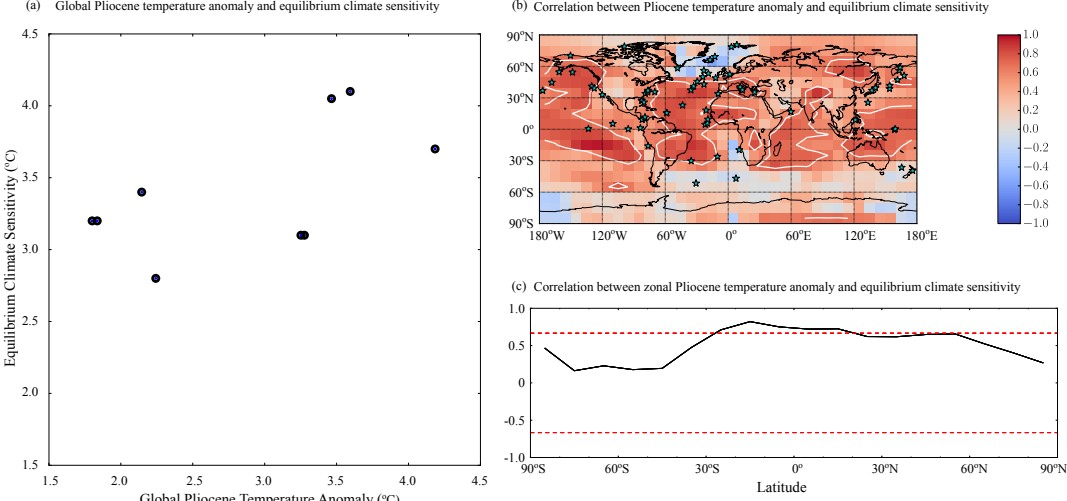

**Figure 1.** Correlations between the PlioMIP anomalies and climate sensitivity. For the Pliocene, the annual SAT anomalies were obtained from averaging the monthly climatology files on the PlioMIP database. For CCSM a 500 year time series is available, so the average over the last 100 years was used. (a) Globally averaged PlioMIP anomaly vs. the estimated equilibrium climate sensitivity from Table 1. (b) The Pliocene temperature anomalies were averaged onto a 10 degree grid and correlated with the global equilibrium sensitivity in each grid box. Cyan stars indicate locations of data points. (c) Zonally averaged results. The dashed lines in plot (c) indicate the 95% significance threshold for a two-sided t-test.

be seen in Figure 1. The simple mean of the anomalies of the 17 data points which lie within the tropical ocean region of all models is rather low at 0.15°C, which is far outside the full ensemble range (taking model values at the same grid points) of $0.9 - -2.4$°C. The spread of data values is also many times greater than the models, at 2.3°Ccompared to 0.15–0.6°Cacross the models (all values 1 standard deviation). In fact, a large majority of the data points lie outside the full range of the model ensemble, in many cases by a substantial margin. Although it is likely that the models do underestimate spatial variation to some extent, it seems reasonable to conclude that much of the model-data discrepancy here is due to uncertainties in the analysis of the data points. Furthermore, we do not expect models to be able to reliably simulate spatial anomaly patterns skilfully at the mPWP, since they fail to do this for other time periods of paleoclimatic interest where sufficient data have been assembled to test this rigorously (Hargreaves et al., 2013). We therefore do not think it is meaningful to constrain the models in this case by a small number of irregularly sampled points, but prefer to focus on averages over larger spatial scales where we can reasonably expect the models to have some skill (Hargreaves and Annan, 2014).

### 3.4 Climate sensitivity estimate

To calculate an estimate for equilibrium climate sensitivity, we combine the model estimates for climate sensitivity and the warming at the mPWP, together with the PRISM3 estimate of tropical ocean temperature change, using the approach described in Hargreaves et al. (2012). For consistency with the data, we use sea surface temperature from the climate models, which are of course very close to SAT at the same locations. The interpolated PRISM3 data indicate a warming of 0.8°C for the SST integrated over 30°S to 30°N. The calculation of climate sensitivity involves sampling from the uncertain temperature distribution, and for each sample, generating a prediction of the associated sensitivity taking account of the uncertainty in the linear relationship. The PRISM3 reconstruction does not include an estimate of uncertainty in the reconstruction. Initially we take a value of 0.4°C (at one standard deviation), based both on the hope that the signal was at least as large as than the noise, and that it might come close to matching the value of 0.7°C (at two standard deviations) which was obtained for a recent reconstruction of the LGM tropics (Annan and Hargreaves, 2013). It is of course essential to test the sensitivity of our result to this assumed uncertainty and we discuss this further below. Figure 2 shows the result. The regression model generates an estimate for the equilibrium climate sensitivity of 1.9–3.7°C. Only the models with weaker tropical warming are consistent with the data, and as these tend to be low sensitivity models, the resulting estimate for $S$ is at the low end of (and extending to values outside) the full range of models that contributed to PlioMIP.

## 4 Uncertainties

### 4.1 Data uncertainty

Proxy-based reconstructions of past climates are, of course, uncertain. As mentioned above, however, the size of the uncertainty in the PlioMIP Experiment 1 SST field has not been objectively estimated and our initial value of 0.4°C is simply an assumption based in part on previous work focussing on the LGM. It would be reasonable to assume that the Pliocene temperature estimate is in fact more uncertain, so we tested the sensitivity of our result to this. The dashed and dot-dashed blue and black lines in Figure 2 show the effect on the estimate of replacing the original estimate of 0.4°C with values of 0.1°C and 1°C (all at one standard deviation) respectively. It is apparent that reducing the uncertainty even to an extremely low value has relatively little effect on the resulting sensitivity estimate (which only narrows marginally to 2.1–3.6°C), as in this case the spread around the regression line makes a dominant contribution to the total uncertainty. However, none of the models are consistent with this temperature estimate, as all warm by more than 0.8°C in the region, many by a substantial margin. If we increase the SST uncertainty estimate substantially to 1°C, then the uncertainty of the overall result does increase more noticeably to 1.3–4.2°C. At this point, even

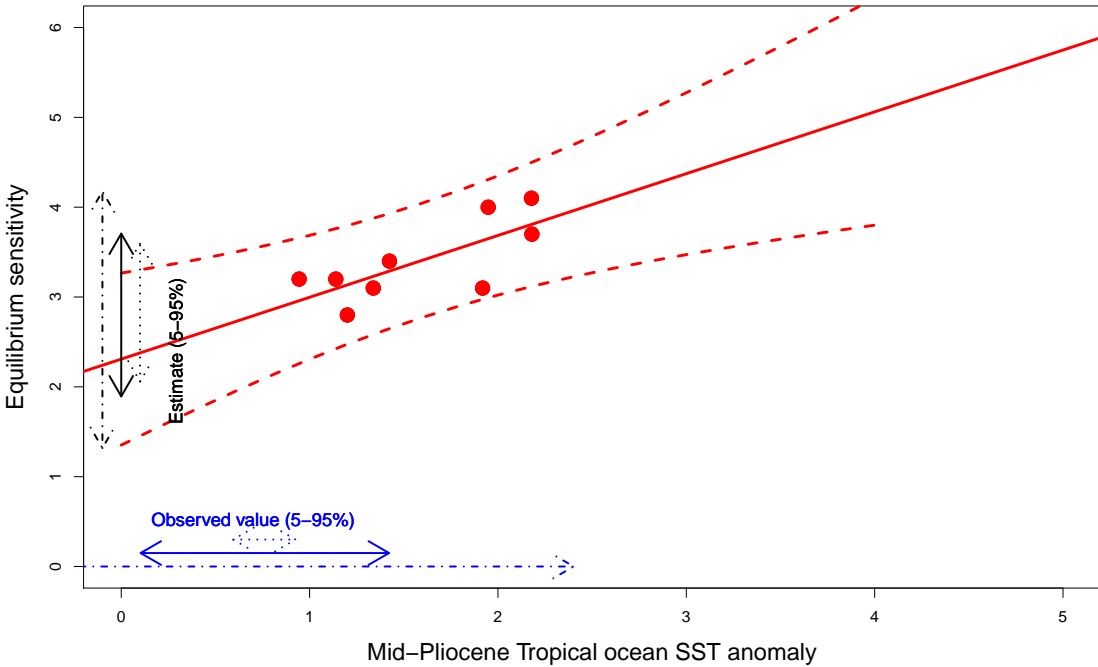

**Figure 2.** Estimating equilibrium climate sensitivity using the mPWP. Red dots represent model values, solid and dashed red lines indicate regression relationship and its uncertainty respectively. Blue arrows show proxy-based reconstruction of tropical temperature change over ocean, together with uncertainty of 0.1 (dashed) 0.4 (solid) and 1.0 (dot-dashed). Black arrows of the corresponding type show the resulting sensitivity estimates.

the models with the strongest warming are consistent with the data and thus the estimated sensitivity range covers the full range of model values with an extension also to lower values. Note that, at this level of uncertainty, the data would no longer give us confidence even that the mPWP was warmer than the pre-industrial, at least in the tropics. It would be very useful to have more complete understanding of the uncertainties of temperature reconstructions for the mPWP.

### 4.2 Forcing uncertainty

A major issue in simulating the mPWP is that the atmospheric $CO_2$ level corresponding to inter-glacial peaks is not precisely known. Furthermore, there is hypothesised to be additional forcing due to methane which cannot be directly inferred from proxy data but which has instead been assumed to be proportional to the $CO_2$ forcing. This was implemented within PlioMIP via an increased

$CO_2$ concentration. That is, the imposed $CO_2$ forcing was selected to represent not only $CO_2$ but the additional effect of methane. Therefore, we have tested the sensitivity of our result to uncertainty in total GHG forcing. Our approach is rather simplistic, and makes the assumption that for each

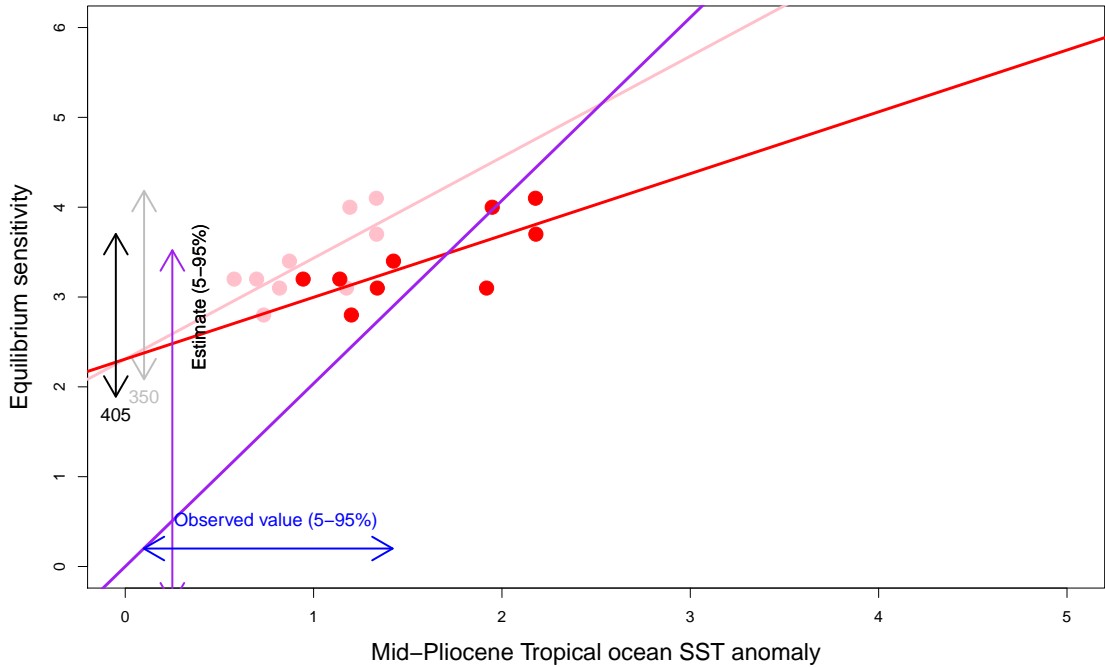

**Figure 3.** Investigating the sensitivity to structural uncertainties. Bold colours show original result, pink and grey show estimated result if 350ppm $CO_2$ were used. Purple shows regression result with zero constant term.

model, the tropical temperature anomaly will change in direct proportion to the net $CO_2$ forcing (relative to the pre-industrial control). While we do not expect this approximation to be precise, it

at least allows us to perform an initial investigation into the sensitivity of our results to changes in the boundary conditions. The PlioMIP protocol imposes a value of 405ppm $CO_2$, but a value as low as 350ppm is possible, being at the low end of the average range considered consistent with the data proxies for $CO_2$ (given as "~360-380ppmv" in Haywood et al. (2010)). When we modify the model results accordingly, we obtain the results shown in Figure 3. By scaling the modelled results

downwards, many more of them are brought into line with the tropical SST estimate derived from the PRISM3 data set, and the resulting sensitivity estimate increases to 2.0–4.0°C. It seems that the value of 350ppm is more consistent with the ensemble as a whole than PlioMIP's own estimate of 405ppm, though of course this cannot be taken to imply that the true value was actually this low.

### 4.3    Modelling uncertainties

The model results are dependent on the experimental protocols, both for the mPWP simulation, and the calculation of $S$. For the calculation of $S$, it is now commonplace to use a regression from an instantaneous 4x$CO_2$ scenario, with this being used in the IPCC AR5 for their model sensitivity

values. However, it is increasingly recognised that this regression-based estimate can significantly underestimate the true equilibrium sensitivity. One of the more extreme examples of this is the GISS model, with the sensitivity reported as 2.1°C in the IPCC AR5 but actually estimated as 2.7–2.9°C by the PlioMIP contributors, based on a long simulation (Schmidt et al., 2014b). For most other models that have done this comparison, the discrepancy is somewhat smaller (Andrews et al., 2015). For the Pliocene experiments, the computational cost of long integrations may mean that some model simulations are not fully equilibrated, which could lead to some errors in their estimates of past and present climates. Internal variability is an additional concern, if the climatology is generated from a short time series. While global temperature is unlikely to be seriously affected by this factor, regional variabilities can be larger. For PlioMIP the intention is that all simulations should be run for at least 500 years, which should produce a reasonably well equilibrated climate, apart from in the deep ocean. However, there may be significant drifts in some regions beyond that time.

### 4.4 Methodological uncertainties

A notable point that is apparent in Figure 2 is that the regression lines do not pass through the origin, but instead indicate that zero tropical warming at the mPWP corresponds to an equilibrium sensitivity of about 2 °C. This may seem a little odd, although it could be argued that even if the response in the tropics was zero, we would still expect a positive response at higher latitudes and thus also in the global average. Additionally, $CO_2$ is not the only forcing in the mPWP experiments (ice sheets and sea level have changed, and vegetation can also change in some if not all models), which does complicate things somewhat. In the LGM analysis, Hargreaves et al. (2012) found that the regression line derived from the PMIP2 ensemble naturally passed close to the origin, so the issue was not apparent concern there. The purple colour in Figure 3 shows the results if we do not include a constant term in the regression. The sensitivity estimate is both lower in its mean value, and much more uncertain, with the 5—95% range reaching from -0.4 to 3.5°C. The increase in uncertainty is due to a combination of there being larger residuals (implying a larger plausible range around the regression line) and also the increased slope of the regression line which means that uncertainty in the true SST anomaly translates into increased uncertainty in the related sensitivity. However, it is worth noting that while the constraint is much weaker, high sensitivity values are still excluded by this approach.

### 4.5 Time slab uncertainties

As mentioned previously, the mPWP data collation is based on averaging the warm peaks within the mPWP interval. However, different locations may encounter peak warmth at different times, and thus the warmest peaks may not represent any historical equilibrium climate state. Moreover, the boundary conditions for the different warm peaks would also have been somewhat different in reality, with differences in orbital forcing potentially leading to regional variation exceeding 1°C (Prescott

et al., 2014) although the variation is lower at larger spatial scales. The comparison between data collected over a wide range of times, and a model snapshot with a specific set of boundary conditions, is thus challenging, though averaging over broad spatial scales should help to isolate any warming signal most clearly. The next iteration of PlioMIP (Haywood et al., 2015) plans to address this issue by focussing on a single interglacial for which sufficient proxy data can be obtained.

### 4.6 Robustness

Robustness of results is a major concern which we have discussed above and summarise here. Caldwell et al. (2014) has highlighted the risk of mining for correlations that are not robust, and there are some examples of plausible correlations in the CMIP3 ensemble which disappeared in CMIP5. Thus we focus on relationships that may be reasonably argued to represent our uncertainties in a realistic manner. In particular, it does not seem at all unreasonable to expect that a greater equilibrium response to increased $CO_2$ in the modern era would also imply a greater response to forcing in the past, and vice-versa, this being a simple expression of the principle of uniformitarianism. Of course in reality the sensitivity depends on underlying climate state and the nature of the forcing (Yoshimori et al., 2011) so the past is not expected to be a perfect analogue of the future, but rather a useful guide. We regard the main result presented here to be a reasonable hypothesis worthy of further investigation, rather than a confident prediction.

## 5 Discussion

The paleoclimate record provides the only observational evidence of large climate changes of comparable magnitude to those anticipated in the coming century. The principle of uniformitarianism implies that the past should be a useful guide to the future. Thus, paleoclimate research forms an important resource of relevance to future climate change. It is, however, not *a priori* clear that any particular paleoclimatic change is immediately informative regarding the future, as the nature of forcings and background climate state may affect the climatic response. Exploration of climate model ensembles provides one route to investigating to what extent a particular past change is in fact informative. The LGM has long been popular as the most recent period in which the climate was substantially different to the present, but as it was colder, large ice sheets were present which complicates the response.

Our results have shown that the mPWP also appears to have some potential for generating useful results. We show there is a strong correlation in the PlioMIP ensemble between tropical temperatures and equilibrium climate sensitivity. Our main result is an estimate for $S$ of 1.9–3.7°C, but major uncertainties in the experimental design and analysis cast substantial doubts over the robustness of this estimate. However, with the evolution of PlioMIP, now moving into phase 2 (Haywood et al., 2015), it seems likely that significant progress can be made on this question in the near future. For

example, the data from the mPWP used here are from a number of different warm periods in the Pliocene which may well represent different climate states (Prescott et al., 2014), and this will be replaced with a more traditional snap-shot of a few thousand years in the next version of PlioMIP. As well as making data more representative of the model simulation, this may also help in establishing an accurate and reliable set of boundary conditions, especially confidence in the level of atmospheric $CO_2$. An improved climate reconstruction would also be helpful; the technology to produce this does exist (Annan and Hargreaves, 2013; Bragg, 2014) and should be applied to the new data set. The small size of the ensemble is clearly a major concern, for which there does not seem to be an easy solution. However, PlioMIP experiments are being included as optional experiments in CMIP6, and the setup is reasonably straightforward even for non-paleoclimate experts to implement, so there are ground for optimism that the ensemble size may increase.

## 6  Data Availability

The PRISM3 SST reconstruction was taken from "Experiment 1 · AGCM version 1.0, Preferred Data", files PRISM3_SST_v1.1.nc and PRISM3_modern_SST.nc, available at the PRISM/PlioMIP webpage, presently located at: http://geology.er.usgs.gov/egpsc/prism/prism_1.23/prism_pliomip_data.html. The PlioMIP model output database was downloaded via sftp from holocene.ggy.bris.ac.uk. Email Alan Haywood (A.M.Haywood@leeds.ac.uk) for username and password.

*Acknowledgements.* This work would not have been possible without the coordinated work of the numerous PRISM and PlioMIP scientists. We are very grateful for the considerable efforts they have gone to to make the fruits of their labours accessible to other scientists. We are also grateful to Dan Lunt and an anonymous reviewer for their helpful comments.

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
