# Peer review of "Could the Pliocene constrain the Equilibrium Climate Sensitivity?"

_Climate of the Past, 2015_

## Short Comment (SC1) · 1 Feb 2016

The concept of using knowledge about conditions during the Pliocene to constrain the equilibrium climate sensitivity (ECS) is attractive. Here the method involves comparing observations of tropical sea surface temperature (SST) warming during the mid-Pliocene Warm Period (mPWP) with that simulated by coupled atmosphere-ocean general circulation models (AOGCMs), and using the relationship in those AOGCMs between the simulated warming and their ECS values to infer an estimated range for ECS in the real climate system.

Unfortunately, there seem to be some serious problems with the results derived in this manuscript. In the light of them, I am not convinced that the results presented can be viewed as robust.
[Figure]

*Lack of a significant relationship when inaccurate model ECS values are corrected*

The method used requires that correct model ECS values are used. The ECS values in Table 1 have, except for FGOALS-g2, been taken from Haywood et al (2013). In two cases, they differ significantly from those given in the appropriate authoritative source, and appear to be wrong.

Haywood et al state the ECS of GISS-E2-R as 2.7–2.9 K; Table 1 and the regression analysis use the mid-point value of 2.8 K. The most authoritative source for the GISS-E2-R ECS is Schmidt et al (2014), which states:

"The climate sensitivity [Charney et al., 1979] for each model configuration is calculated using the q-flux model (with a maximum mixed layer depth of 65 m to reduce computation time) to estimate the climate response to 2xCO2. The NINT, TCAD and TCADI ModelE2 versions have sensitivities of 2.7°C, 2.7°C, and 2.9°C, respectively... Estimates of the coupled ocean model sensitivity ... are 2.3°C, 2.3°C, and 2.4°C and 2.5°C, 2.4°C, and 2.5°C, respectively for the three physics-versions and the two ocean models (R and H)."

It appears that Haywood et al, who cite the submitted version of Schmidt et al. as their reference for GISS-E2-R, have confused the ECS of the mixed layer version of GISS ModelE2 with that of the coupled model. The standard version of GISS-E2-R is NINT; there is no indication that the TCAD or TCADI version was used for the PlioMIP experiments. Accordingly, the relevant ECS value per Schmidt et al. 2014, which should be used for this study, is 2.3 K. Note that Table 9.5 of AR5 WG1 gives the estimated ECS of GISS-E2-R as 2.1 K, but this is derived using the standard estimation method of regressing over the first 150 years of the abrupt4xCO2 CMIP5 experiment. According to Gavin Schmidt's presentation at Ringberg2015, for GISS-E2-R this regression method underestimates its true ECS value of 2.3 K.

Also, Table 1 and the regression use the ECS value for the IPSL-CM5A model given by Haywood et al of 3.4 K. However, Dufresne J L et al 2013, the reference cited by

Haywood, states:

"While the climate sensitivity of IPSL-CM5A-LR ($\Delta T_s{}^e$(2CO2)  4.1K) lies in the upper part of the sensitivity range of the CMIP3 models, the sensitivity of IPSL-CM5B-LR ($\Delta T_s{}^e$(2CO2)  2.6K) falls in the lower part".

The 4.1 K ECS value for IPSL-CM5A is supported by the abrupt4xCO2-based regression lines in their Figure 24a, and the related values of 4.10 K for IPSL-CM5A-LR and 4.12 K for IPSL-CM5A-MR given in their Table 1. Note that Dufresne et al. state in their conclusions that "The equilibrium climate sensitivity of IPSL-CM5A and IPSLCM5B are drastically different: 3.9 and 2.4 K, respectively.", which is not quite the same. It appears that, in the final published paper, that wording was inadvertently not updated from the wording in the same sentence in the conclusions section of the submitted manuscript, to reflect the revised values given in the first sentence quoted and in their Table 1. In the submitted manuscript, the values appearing in the first sentence quoted, earlier in their text, and in their Table 1, were 3.9 K and 2.4 K.

Hourdin et al (2013), discussing the IPSL-CM5B model, likewise state that: "The climate sensitivity of the new IPSL-CM5B model is thus much smaller than that of the previous IPSL-CM5A model. For a doubling of CO2, the temperature increase is approximately half of that for a quadrupling of CO2, i.e. around 2.7 K for IPSL-CM5B and 4 for IPSL-CM5A".

The estimated ECS for IPSL-CM5A in AR5 WG1 Table 9.5 is 4.1 K.

Clearly, 3.4 K is not the correct ECS value for IPSL-CM5A; the most authoritative value is 4.1 K, from Dufresne et al (2013).

So far as I can tell, the remaining CMIP5 model ECS values are close to their most authoritative estimates.

Unfortunately, when the regression is re-run with the correct ECS estimates for the two models with incorrect values, 2.3 K for GISS-E2-R and 4.1 K for IPSL-CM5A, the

tropical SST – ECS relationship is no longer significant even at the 5% level. Using accurately digitised tropical SST changes per Figure 2, the correlation drops from 0.75 (on my calculation) to 0.59 (regression R2 0.35), with a p value of 9.3% rather than 1.9% using the uncorrected ECS values.

The lack of a statistically-significant relationship using the chosen regression model, when corrected values for model ECS values are used, appears fatal to drawing valid conclusions about ECS based on this model, even if there were no other issues with the regression model and approach used.

*Realism of model mPWP simulations and extrapolation*

The authors state (line 49) the rationale underlying the method as: "if the past behaviour of the models is indicative of their future behaviour in some relevant manner, then it should be possible in principle to use observations of the past to deduce which models are more reliable and hence generate a constrained forecast of the future."

Even if the regression relationship were strong, I don't think that the rationale underlying the method applies in this case. The idea logically involves ascertaining which models generate realistic results, on the basis of the tropical SST change in their mPWP simulations, before using the regression relationship to deduce what range of model ECS values corresponds to the uncertainty range for observed SST change. In the case of mPWP tropical SST change, most models lie outside the observed range, with one at its 80% point, two around its 92.5% point and the other six above its 95% point. That implies a strong probability that almost all, if not all, of the models do not generate realistic simulations of tropical SST change for the mPWP. Why, then, should one regard the relationship between tropical SST warming and ECS in models, almost all of which generate unrealistic warming, as reliable?

This problem is acute here because the relationship is extrapolated well beyond the area occupied by the models; the 5% point of the observed values corresponds to SST warming of only 0.04 times as much as that generated by the model with the

lowest warming. Even the 50% point is only 0.68 times as high as the warming in that model. I do not see how extrapolating from a relationship amongst models that almost all generate unrealistic results can produce a realistic uncertainty range.

*Regression with intercept methodology*

In many cases where an emergent constraint approach is used the observations involved may have no direct connection to the characteristic of the model that is of concern. However, in this case the observation is of the tropical SST change arising in the mPWP, attributed primarily to greenhouse gas forcing (albeit with slow as well as fast feedbacks having occurred). There seems good reason to expect a direct, if imperfect, relationship between tropical SST change and ECS in models. If a model has an ECS of zero, the SST change should be approximately zero. The slope of the ECS–SST relationship will vary between models, reflecting inter alia differing ratios of Earth system sensitivity (ESS) to ECS and of tropical to global surface temperature changes. Nevertheless, if the relationship between model ECS and mPWP tropical SST change is informative about model ECS then would one not expect the regression line to pass within a reasonable distance of the origin? It could be argued that regressing with no intercept term may be more consistent with the physical relationships involved. The fact that the best fit line when regressing with the intercept as a free parameter crosses the zero SST change line at an ECS of over 2 K is certainly of major concern. If regression with no intercept term were used, the median ECS estimate would be 50% lower, so this issue is of first order importance. Although the issue is discussed briefly in the manuscript (section 4.4), it is not in my view adequately addressed.

Perhaps a less unsatisfactory method would be to scale each model's ECS value in proportion to the ratio of its simulated mPWP tropical SST change to the (probabilistic) observed value, generating for each model a probability density for a scaling-based estimated ECS. One would then average these PDFs over all models, perhaps down-weighting those with the least realistic simulations, or use some other method of combining the individual model scaled ECS estimates. This approach would recognise that
models exhibit different relationships between ECS and mPWP tropical SST change.

*Robustness of a relationship found amongst a group of AOGCMs*

When a similar approach to that used in this manuscript was used in Hargreaves et al (2012), in relation to Last Glacial Maximum (LGM) simulations by CMIP3 models, the issues identified in this comment were either inapplicable or of minor importance compared with in this case. Nevertheless, the strong relationship that they found in CMIP3 AOGCMs between tropical warming post the LGM and ECS did not survive into the CMIP5 generation of models (Hopcroft and Valdes 2015). That must make for caution in the much more problematic mPWP case. The robustness issue is discussed in section 4.6 of the manuscript, but I am not sure that the caveats given sufficiently reflect the great fragility of conclusions drawn in this case, where almost none of the models produce realistic simulations of mPWP tropical warming.

References

Dufresne, J. L et al. (2013). Climate change projections using the IPSL-CM5 Earth System Model: from CMIP3 to CMIP5, Clim. Dynam., 40, 2123–2165.

Hargreaves J C et al (2012). Can the Last Glacial Maximum constrain climate sensitivity? Geophys. Res. Lett., 39, L24702, doi:10.1029/2012GL053872.

Haywood, A. M.et al. (2013). Large-scale features of Pliocene climate: results from the Pliocene Model Intercomparison Project, Climate Of The Past, 9, 191–209.

Hopcroft, P. O., and P. J. Valdes (2015). How well do simulated last glacial maximum tropical temperatures constrain equilibrium climate sensitivity? Geophys. Res. Lett., 42, 5533–5539, doi:10.1002/2015GL064903.

Hourdin F. et al. (2013): Climate and sensitivity of the IPSL-CM5A coupled model: impact of the LMDZ atmospheric grid configuration, Clim. Dynam., online first: doi:10.1007/s00382-012-1411-3, 2012.

---

## Referee Comment (RC1) · Anonymous Referee #1 · 7 Feb 2016

This paper derives a relationship from an ensemble of AOGCM simulations of the mid-Pliocene between tropical temperature difference wrt present (Tp) and equilibrium climate sensitivity (S, ECS), and applies it to an estimate of Pliocene tropical SST to derive bounds for ECS. I think that it is useful to attempt to do this in principle, but I am not convinced by some aspects of the method. Consequently I am not confident of the results obtained.

General comments

My first two comments are reservations about the relationship at line 90

S = alpha Tp + C + epsilon

which is fitted by linear regression of the models.

[Figure]

(1) Firstly, no comment is made at line 90 about C, and it later turns out that C is substantial compared with S. I would expect that if ECS is zero, both global mean and tropical temperature change wrt present will be near zero. Tp might not be quite zero because there is unforced variability in any simulation, and there could also be local effects of heterogeneous forcing. The general idea of climate sensitivity is that forcings cause a global response. Many experiments with GCMs show that the pattern of response in a given model is fairly constant while the amplitude depends on the magnitude of the forcing. I am sure the authors know this. In sect 4.4 they briefly discuss this problem. They suggest that the tropical temperature change could be zero for a non-zero ECS. But that contradicts their own expectation at line 67, where they note that in LGM simulations there is large Tp with large S. The latter is also suggested by Fig 1, where the tropical response is greater than the global mean. If Tp were near zero it would not contain any information that could constrain ECS i.e. it invalidates the assumption of the method. If C is omitted, the line is constrained to pass through the origin, and the conclusions will be substantially modified.

(2) Secondly, the authors argue that S should be treated as the dependent variable and Tp the independent in a regression. This seems surprising. In an ensemble of models simulating climate change, the T change in a small region will generally have a larger fractional spread than the global T change. Because of this, and because ECS refers to a global energy balance that determines global T change, I think it would be more natural to make S the independent variable. Alternatively TLS could be used, as the authors say, given an independent estimate of the ratio of the uncertainties in S and Tp. However, I would suggest that the treatment of the scatter entirely as an additive epsilon is not appropriate anyway. As I said in the first point, many results show that a given GCM tends to have a fairly constant pattern of T change, but the pattern is model-dependent. Thus, we might expect the ratio of tropical to global T change to be a model-dependent ratio R, and $Tp = S\,R\,F/F_{2x}$, where F is the Pliocene forcing and $F_{2x}$ the $2xCO_2$ forcing used to define S. There is probably additive noise as well, but at least part of the scatter in Tp versus S is due to R, which is a multiplicative factor. I

suggest that the effects of the spread of R and S in the PlioMIP ensemble should be considered separately.

(3) I am also concerned that Tp in the models appears to be systematically larger than the proxy estimate. The authors comment on this in Sect 4.1. Isn't this a serious problem? It might mean that the proxy data is wrong, the BCs used for the AOGCM experiments are wrong, or that the models are wrong (they might produce the wrong R, for example). In any of these cases, the method is compromised.

(4) The treatment of the uncertainty of the Pliocene tropical SST estimate seems inadequate. The authors have derived a tropical-mean annual-mean by interpolation and integration from the proxy dataset, which was presumably rather sparse initially as well having uncertainties on the data. I don't find it satisfactory to state simply that the uncertainties are not known and therefore make some fairly arbitrary choices, since the final result depends substantially on this.

Specific comments

13. I don't think it helps to call it "Charney". I would recommend omitting that. If the authors mean that certain things are included and others are not, it would be better to spell them out. Throughout the text and figures, I would recommend using the phrase "equilibrium climate sensitivity" (or ECS). The phrase "climate sensitivity" alone is rather vague. It might mean the climate sensitivity parameter (in K per W m-2).

24. Why do the ice sheets particularly make it a challenge? Ice-sheets give a global forcing which can be taken into account in estimating ECS.

30. Similarly, why is it an advantage that CO2 was higher in the mid-Pliocene? I am not arguing against this or the previous point, but I think they should be justified.

33. More is needed here to explain what ESS is and why it is different from ECS, so that the reader can be clear what is new about the present paper.

52. Should it be "constrain" rather than "predict"?

91. I think alpha is not completely unknown as it should contain the ratio of the forcings.

98. Are the two PlioMIP experiments for the same climate conditions? They have to be consistent, because of the use of the SST from Experiment 1.

109, 225. If the AOGCMs are not run to equilibrium, the ratio of Tp to global T might not be characteristic of equilibrium. The suppressed warming in the Southern Ocean in Fig 1 might indicate they are not equilibrium. This could bias the results.

112, 124, 200. It would be useful to quantify the various forcings, so that we can appreciate how they compare with the $CO_2$ forcing. Either $CO_2$ has to be overwhelmingly dominant, or we assume that climate sensitivity is the same for all forcings (that's the usual assumption, although not completely accurate, as the authors later note).

133. Please quantify the correlation and test its significance.

139. What is the precise definition of "the tropical region"?

217. Please give some references for this "commonplace" method.

219. Increasingly recognised by whom? References please.

223. Andrews et al. use abrupt4xCO2, not 1pctCO2.

227. Although the surface temperature trend may be small after 500 years, it can go on for a long time, and thus global T change may be substantially short of its eventual value. See e.g. Li et al (2012, 10.1007/s00382-012-1350-z).

256. What is uniformitarianism?

257. There are many earlier references for this e.g. Joshi et al. (2003, Clim Dyn), and more recently e.g. Shindell (2014, Nature Climate Change).

---

## Author Comment (AC1) · 18 Feb 2016

At the outset, we aimed to explore the outputs of PlioMIP to investigate how best to use these. It is well known that the variety of slightly different definitions of what is meant by equilibrium sensitivity can give rise to different answers. This ambiguity is an additional source of uncertainty that could have been discussed more clearly in our manuscript, although we do not think it has a substantial influence on our results as the different approaches generally give similar results. In the two cases noted: for IPSLCM5A, the value of 3.4C quoted in Haywood et al appears to match closely to the $2\times CO_2$ value of 3.47C presented in Table 1 of Dufresne et al (whether the small difference may be due to a rounding error, or some other source of variation is not clear). Thus, we see no basis for changing this value. As for GISS, we were aware of the range of values that had been generated, and discussed this with the relevant co-authors on the Haywood

paper. We are happy to confirm that the value presented in Haywood et al was in fact the deliberate choice of these authors. However it is arguable that the lower value of 2.3 would have been more appropriate, depending in part on what method other authors used for their sensitivity values (which is not always clear in the literature). We did test changing to the lower value, and it made very little difference to our result with correlation still being significant at the 95% level. Thus we didn't think it appropriate to over-rule the choice made in Haywood et al. In fact, even when changing both the GISS values to 2.3 and the IPSL value to 4.1 as suggested in your comment, the regression is still significant at the 5% level, contrary to your assertion.

We agree that it is a concern that the observations are towards the lower end of the model range. However, when uncertainties are considered, there is substantial overlap. There are reasons to believe that the MPWP forcing may be on the high side, and changing this would improve the match, as discussed in the manuscript. We are certainly not concerned that the models do not cover the full range of uncertainty allowed by the observational analysis. GCMs are constructed so as to obey a large number of physical principles and (unlike simple energy balance models) their sensitivity is an emergent property that cannot be arbitrarily selected. There may be very good reasons why no-one has yet built a reasonable climate model with a sensitivity much less than 2C, for example. Given the uncertainties in proxy interpretation and forcing, we would be reluctant at this point to confidently assert that most models have warmed unrealistically. Of course, all models are inevitably wrong and each one will either be too warm or too cold compared to reality. The purpose of the regression is not to select which model or models are "correct" but rather to estimate where on the y-axis a perfect model could be expected to lie, if the relationship found across the existing ensemble holds. Our choice of conditional form "could" in the title of the paper, and caveats stated throughout the manuscript, were quite deliberate.

Regarding the non-zero intercept, we agree this is an interesting issue and thank you for raising it. We agree that it would be natural to have anticipated a priori that the

regression would pass close to zero, and this does raise the issue of whether it should be constrained to do so. On the occasions that we have presented this work, we have discussed this issue with other climate scientists, but have not arrived at a clear physical explanation for the non-zero intercept. Thus we think that it would be useful to include further discussion and the calculation as an alternative result and plan to do so in the revised manuscript.

———————————————————

---

## Author Comment (AC2) · 18 Feb 2016

Thank you for the comments. Here we discuss some of the major points raised.

As a general comment: We certainly share some of the reservations that the reviewer has with the results. There are several challenges with obtaining a robust and credible result, and this analysis with the currently available models and data is intended as a step towards this, rather than a final answer. We did intend this viewpoint to come across clearly in our manuscript (starting with the title) but will try to address this more unambiguously in the revision.

Non-zero intercept: Yes, it is an interesting point as to whether such a substantial intercept is plausible. There are, as discussed in the manuscript, reasons why the line might not necessarily be expected to pass precisely through zero. Natural variability

cannot explain such a large value but a hypothetical model that is globally insensitive to $CO_2$ would not necessarily be regionally insensitive to $CO_2$ let alone different forcings. For example if such a model were to achieve zero mean global temperature change under $CO_2$ forcing by cooling at high latitudes to offset tropical warming, then a warm tropics in the MPWP simulation would be expected. Such thought experiments may not be intuitively comfortable, but we think that is largely because the concept of a model with zero sensitivity to $CO_2$ is hard to swallow. The extrapolation to zero is also quite a way outside the model range such that the linear approximation may fail. However, it is definitely an interesting question as to why the models with lower climate sensitivities generate similar tropical MPWP warming to the models with substantially higher sensitivities. In the revised manuscript, we plan to additionally present results in which the regression is performed with the constraint of a zero intercept.

As for the nature of the regression: We do not see why the larger fractional spread should determine the direction of the regression. It is certainly simple to generate artificial examples where this is not the case, as the direction of the regression depends on whether the residuals are independent of one variable or the other as discussed in the paper. For any regression on data (Xi,Yi) it is always possible to define the ratio Yi/Xi for each pair of points, but this does not tell us anything about the regression residuals. We do not believe that residuals would necessarily be zero for zero sensitivity models, which would make multiplicative errors inappropriate.

It is indeed notable that the estimate arising from the data lies very much at the end of the model range, though clearly overlapping it when realistic uncertainty is taken into account. One possible explanation of this is that the forcing due to boundary conditions (particularly $CO_2$) may be excessive - a value as low as 350ppm $CO_2$ is also regarded as quite plausible and would result in a much better agreement between models and data (according to our simple adjustment presented in the manuscript). Additionally, the data used in PlioMIP do not represent a true time slice, which makes it hard to be confident in a direct comparison with model simulations, even assuming that the

calibration of the data points was robust (which history suggests may not be the case). The next iteration of PlioMIP, focussing as it will do on a specific time slice within the MPWP, should provide clearer results.

As for the treatment of the uncertainty in the Pliocene SST data, we would be open to any helpful suggestions. We didn't do any processing on the data ourselves, but simply used what has been published, for which no uncertainties have been provided. We have discussed this with some of the scientists responsible and decided that the most appropriate way forward was for us to at least test the sensitivity of our results to a plausible range of uncertainty values. We hope that the data analysis will be more quantitative in future, but the current state of the art is as reported. We think it is useful to investigate the sensitivity of the results to a reasonable range of uncertainties on the data.

We will reply to the detailed comments in the formal response to reviewers.

---

## Short Comment (SC2) · 10 Mar 2016

Just a comment- I don't think you should be using the interpolated values for the PRISM3 SST field. Whatever confidence we can put on the SST's only applies to the sites themselves.

Also, while the PRISM3 SST fields are being used in your analysis, you fail to provide the correct citation. It is not the PlioMIP experimental design paper. The correct reference for SST is:

Dowsett, H.J., Robinson, M.M. and Foley, K.M., 2009. Pliocene three-dimensional global ocean temperature reconstruction. Climate of the Past, 5(4): 769-783.

and for the entire PRISM3 reconstruction:

[Figure]

Dowsett, H.J., Robinson, M.M., Haywood, A.M., Salzmann, U., Hill, D., Sohl, L., Chandler, M., Williams, M., Foley, K. and Stoll, D.K., 2010. The PRISM3D paleoenvironmental reconstruction. Stratigraphy, 7(2-3): 123-139.
* * *

---

## Referee Comment (RC2) · D. Lunt (Referee) · 11 Mar 2016

First of all, apologies for the delay in providing this review.

Secondly, although I glanced briefly at the comment of Nicholas Lewis, I have not read the other reviews or comments, and so my review below can be regarded as independent.

This paper provides an interesting analysis of whether Pliocene and high-$CO_2$ modelling combined with Pliocene data can constrain climate sensitivity. The methodological approach seems sensible and well justified.

My main comment is that the range given in the Abstract of 1.8-3.6°C is misleading in my opinion. This is the range given by the 'first attempt', but subsequently additional

factors are considered in the paper which better reflect the true range of uncertainty. These should also be added to the Abstract. Also line 274, I disagree that the "main" result is 1.8-3.6°C. This is an initial result that you then demonstrate to be an underestimate of the range.

Specific comments:

(1) The abstract needs to give a brief description of the methodology.

(2) Line 28. Could also add that in the Pliocene the anomaly goes in the same direction (i.e. warming) as a doubling of $CO_2$, and so is potentially more appropriate than the LGM for constraining climate sensitivity.

(3) Line 31. Could also add that the Pliocene is an appropriate time period because there has been a huge effort to generate a large observational dataset, mostly by the USGS group of Dowsett.

(4) In the title and abstract and throughout, I am not sure exactly what 'constrain' really means. If the Pliocene could show that climate sensitivity was somewhere between 0 and 10°C, would that be a 'constraint'? I think so. As such, the question posed in the title is rather trivial. I think the real question is 'By how much' could the Pliocene constrain climate sensitivity. Or, do you mean 'Could the Pliocene constrain climate sensitivity to a range narrower than the currently accepted range'?

(5) You state that the Pliocene simulations have probably been carried out for different lengths of time. As such they are at different degrees of equilibrium. This could be partially addressed by accessing the timeseries of temperature in the PlioMIP database and extrapolating the shortest simulations using e.g. curve-fitting to the timeseries, or Gregory plots if the necessary data is available. The lengths of each simulation could easily be obtained if you asked!

(6) Lines 78-95. I confess to not following this section, or even understanding what the issue being addressed is. Intuitively I would think that it didn't matter whether the graph

in e.g. figure 2 was plotted as x-y or as y-x.

(7) Table 1. It is not clear to me that these numbers are completely comparable with the Pliocene equivalents. (i) For the same reason as (5) above – the simulations have probably all been run for different lengths, and (ii) because they may have been run with different versions of the model. Again, this needs to be highlighted and discussed. It might be a good recommendation form this paper that for PlioMIP2 a high-$CO_2$ simulation is carried out explicitly with exactly the same model and for the same length of time.

(8) I would strongly recommend that you only use the raw point-wise data ( 70 sites I think) of the PRISM dataset for this analysis. The gridded data used in this paper for evaluation was completely made-up in locations which are far from the raw data (no offence to PRISM!).

(9) Line 160. This seems a little lazy. Modelled SSTs are also in the PlioMIP database, so you should these rather than SATs to compare with the PRISM SSTs.

(10) Line 168. Taking $0.4°C$ as the initial uncertainty on the data seems over-optimistic to me. I would start with $1°C$ which is the number often used, and then increase to 2, or even $3°C$ in the sensitivity studies.

(11) Line 214. Note that the $CO_2$ proxies only give us $CO_2$, but there may well be changes to non-$CO_2$ greenhouse gases, hence why PlioMIP prescribes a $CO_2$ level greater than many of the proxies.

(12) Line 231. I don't think it's necessarily surprising that zero tropical warming gives a non-zero sensitivity to CO2. Some of the forcings in the Pliocene that given global mean warming (low Rockies, reduced ice sheets) have zero expression in the tropics. See e.g. Lunt et al (2012), Figure 4.

(13) Line 284. Actually, Bragg (2014) did apply the methodology to the Pliocene, and is included in the reference given.

(14) I like the 'data availability' section! Could add where you got the CMIP5 numbers from in Table 1.

Technical comments:

(A) Line 37+38. 'greater' rather than 'higher'?

(B) Line 210. "downscaling" is often used in a different context. Maybe "adjusting appropriately".

Lunt, D.J, Haywood, A.M., Schmidt, G.A., Salzmann, U., Valdes, P.J., Dowsett, D.J., Loptson, C.A. (2012). On the causes of mid-Pliocene warmth and polar amplification. Earth and Planetary Science Letters, 321-322, 128-138.

---

## Author Comment (AC3) · 27 Apr 2016

Here we provide additional comments in reply to anonymous review 1, including detailed responses to the minor points.

Further to the discussion on the regression, we will also include the regression with zero constant term and include this fully in our discussion. As the reviewer predicts, this does of course generate a rather different result, extending to much lower values while still reaching similarly high values to the original result.

Also, please note that changing to modelled SST rather than SAT has clearly improved the agreement between models and data, though without materially affecting the main results. This is due to the combined effects of both some changes in the anomaly values, together with, together with minor changes to the gridding and masking. The

observational estimate now overlaps more substantially with the range of model results. Nevertheless, there does still appear to be a general bias between models and data.

Minor points:

Charley sensitivity is quite widely used in the literature (often in contrast to Earth System sensitivity) and we think it's worth clarifying that this is what we are using exclusively in the paper. We have changed to ECS in multiple places.

24 Ice sheet forcing is not known precisely, and perhaps more importantly, it does not add linearly with other forcings (in GCMs where this can be, and has been tested).

30 The argument is simply that the response will not be precisely linear over a large range of positive and negative forcings (even restricted to CO2) and hence the historical response to high CO2 is the scenario which we may expect to give the best estimate of the future response to high CO2.

33 Agreed, we will expand this section

52 yes probably

98 yes they are attempting to simulate the same climate, with AGCM (prescribed SST) and AOGCM.

109,225 Agreed and we will explicitly mention this

133 Will do

139 30S-30N, now included

217,219,223,227 We will make some minor edits here

256 We use this term to refer to the underlying principle of constancy in natural laws, which underpins the concept of repeatability in experiments

257 Noted.

---

## Author Comment (AC4) · 27 Apr 2016

Thank you for the comment. We will add the correct reference to the revised manuscript. We prefer to use broad-scale averages as they should provide less noisy estimates of temperature changes, and be less susceptible to location errors in the models. A full spatial reanalysis was outside the scope of this work so we used the PRISM3 analysis as a plausible first step. We certainly look forward to any future improvements in the data and may ourselves attempt a full spatial reconstruction along the lines of our work with the Last Glacial Maximum (Annan, J. D. and Hargreaves, J. C.: A new global reconstruction of temperature changes at the Last Glacial Maximum, Climate of the Past, 9, 367–376). But this is left for future work.
* * *

---

## Author Comment (AC5) · 27 Apr 2016

Thank you for the helpful review.

We will reconsider the wording in the abstract and the main text in order to ensure that readers don't get a misleading impression of our results. We certainly don't want the headline value to be taken as definitive.

As for the specific comments:

1 We will improve this aspect of the abstract.

2 Yes, we will emphasise this point (also relevant to rev1).

3 A good point and we will add the Dowsett reference.

[Figure]

4 It is perhaps best to not over-interpreting the term "constraint". However, if it can be argued that the Pliocene constraint is independent of others, it may still be useful even if weak (e.g. Annan and Hargreaves 2006).

5 We did look at this, but were not able to reach any firm conclusions. It is not easy to separate out internal variability (on long time scales) from drift, especially when looking regionally. So we can only mention it as a caveat and not provide a concrete answer.

6 It does matter. Reference will be added and section improved.

7 We have improved the discussion of the various inconsistencies in calculations of sensitivity. For the most part, however, it seems likely that these uncertainties are not large compared to the inter-model differences. It also appears that CMIP6 has given up on the calculation of equilibrium climate sensitivity via long equilibrium runs, pre-ferring the related but distinct concept of effective sensitivity using regression following an abrupt change in forcing ("Gregory method"). PlioMIP may prefer to follow this approach.

8 This is an interesting point and one where we respectfully disagree with the reviewer. We do consider it strongly preferable to use averages over large areas, to maximise the relative importance of the large-scale forcing and minimise the influence of loca-tion errors. We do acknowledge your concerns about the reliability of the PRISM3 reconstruction and hope to see (and perhaps contribute to) more robust and traceable analyses in future iterations of the project. The main goal of this work was to see if such research is worth pursuing, and as such the numerical result itself (which we agree has limited credibility) is of secondary importance. Furthermore, dealing with individual data points would introduce the complication that we would have to make some additional assumptions regarding the covariance of errors across multiple sites. This would require a level of data analysis and interpretation well beyond the scope of this research.

9 We have re-done all the calculations with SST. This does clearly improve the agreement between models and data, due to the combination of slightly different anomalies, and some changes to the gridding and masking. However, there is still a substantial mean bias between models and data.

10 Note that this value is the uncertainty of the area mean, not that of the individual points. We can reasonably hope that the former will be considerably lower than the latter, though we are not yet in a position to evaluate this in detail.

11 Noted, and will be mentioned.

12 Noted and these forcings will be specifically mentioned. However, the issue is the other way around: according to our result, a model with zero (global) sensitivity will still warm in the tropics.

13 Noted.

14 Will do.

Technical comments

a,b Will do.

---

## Author Response (AR1)

**Reply to Editor:**

We have looked at the individual data points. Please see the new discussion in Section 3.3 of the manuscript and the response to point 8 in Dan Lunt's review. Thanks for reminding us of the Prescott et al work which provides a useful indication of the limitation of the "time slab" approach, and further supports the use of large scale averages (since the temperature variation is reduced at larger spatial scales).

Please find the manuscript diff file below the responses to the reviewers.

**Dowsett:**

Reference now included (Sect 3.3). We have looked at the pointwise data but can say little more than that they are incompatible with the models.

**Ref#1:**

(1,2,3) The choices made in the regression model are discussed more fully in Section 2, and we now also include the regression with zero constant term, which is shown in Fig 3. Now we are using SST, the agreement between models and data is slightly improved.
(4) We agree the data are problematic. We are not in a position to derive uncertainties ourselves as part of this research, but hope that this work will provide additional motivation for progress towards this in the future.

We have attended to all the minor comments and a number of changes have been made to the text:

Charney sensitivity is quite widely used in the literature (often in contrast to Earth System sensitivity) and we think it's worth clarifying that this is what we are using exclusively in the paper. We have changed to ECS in multiple places.

24,30,33,52 text amended.

Possibly true, but as this does not influence our analysis, no change made.

Responded to in author's comment. No change made.

(109) Text in Section 4.3 has been changed to reflect this concern.

(112,200) comment and reference added.

correlation values more fully stated.

Tropics is now stated as 30s-30N.

217-223 this was a bit garbled in the original and has been rewritten.

256. Responded to in author's comment. No change made.

257. Assuming I have found the correct papers, they seem to relate to the forcing only, and do not include the paleoclimate context (and one is actually a later reference). No change made.

**Lunt:**

(main comment and specific comment 1) We have changed the abstract to emphasise the uncertainties more clearly, and also to briefly introduce the methodology (2,3) done (5) discussed more clearly in the text.

(6) additional discussion and explanation, and reference added (7) CMIP6 has rejected the idea of long equilibrium 2xCO2, so PlioMIP is probably best advised to be compatible with them, though this isn't really our business.

(8) We've looked at the data and added a discussion of our analysis to section 3.3. The position of the points where we have an estimate of the annual mean temperature anomaly from the data have been added to the map in Figure 1. Taking the data on a point by point basis we can't really form any conclusions at all (other than that the data is incompatible with the models and not infrequently with itself). However, our previous research emphasises the benefits - and even necessity - of using large scale averaging for useful model data comparison, as even if the data are good, it's been repeatedly shown that models cannot represent small scale patterns accurately.

You stated, "The gridded data used in this paper for evaluation was completely made-up in locations which are far from the raw data (no offence to PRISM!)." However, from reading the description in Haywood et al. (GMD, Expt1, 2010) and also Dowsett et al it seems to me that the PRISM SST anomaly is close to being a scaling of the present day temperature pattern. While clearly not perfect, this is not a wholly unreasonable first-order estimate.

(9) We have redone all the calculations with SST, which hardly changes anything other than perceptibly improving model-data compatibility.

(10) We use a range of small and large values to indicate how sensitive the results are to this factor. Perhaps 0.4C would be best regarded as a hope for the future than a current estimate.

(11-13) Minor adjustments made to text (14) As stated in the caption, the values for sensitivity estimates are from Haywood et al.

A: stronger!

B: wording change here

[revised manuscript text omitted]